# Clinical Progression of *Theileria haneyi* in Splenectomized Horses Reveals Decreased Virulence Compared to *Theileria equi*

**DOI:** 10.3390/pathogens11020254

**Published:** 2022-02-16

**Authors:** Kelly P. Sears, Donald P. Knowles, Lindsay M. Fry

**Affiliations:** 1Department of Clinical Science, Carlson College of Veterinary Medicine, Oregon State University, Corvallis, OR 97331-4801, USA; kelly.sears@oregonstate.edu; 2Department of Veterinary Microbiology and Pathology, Washington State University, College of Veterinary Medicine, Pullman, WA 99164-6630, USA; dknowles@wsu.edu; 3USDA—ARS, Animal Disease Research Unit, Pullman, WA 99164-6630, USA

**Keywords:** *Theileria haneyi*, *Theileria equi*, virulence, splenectomy

## Abstract

The global importance of the hemoparasite *Theileria haneyi* to equine health was recently shown by its resistance to imidocarb dipropionate (ID) and its interference with *T. equi* clearance by ID in some co-infected horses. Genetic characterization of *T. haneyi* revealed marked genomic reduction compared to *T. equi*, and initial experiments demonstrated reduced clinical severity in spleen-intact horses. Furthermore, in early experiments, splenectomized horses survived *T. haneyi* infection and progressed to an asymptomatic carrier state, in stark contrast to the high fatality rate of *T. equi* in splenectomized horses. Thus, we hypothesized that *T. haneyi* is less virulent than *T. equi*. To objectively assess virulence, clinical data from nine splenectomized, *T. haneyi*-infected horses were evaluated and compared to published data on *T. equi*-infected, splenectomized horses. Seven of eight splenectomized, *T. haneyi*-infected horses survived. Further, in six horses co-infected with *T. equi* and *T. haneyi*, only horses cleared of *T. equi* by ID survived splenectomy and became asymptomatic carriers. The reduced virulence of *T. haneyi* in splenectomized horses instructs why *T. haneyi* was, until recently, undetected. This naturally occurring comparative reduction in virulence in a natural host provides a foundation for defining virulence mechanisms of theileriosis and Apicomplexa in general.

## 1. Introduction

*Theileria equi* and *Theileria haneyi* are closely related Apicomplexan hemoparasites and known causative agents of equine theileriosis. *T. equi* has been detected in tropical and subtropical regions around the world, and *T. haneyi* has been documented on multiple continents, with its distribution anticipated to expand as surveillance for the organism increases [1,2,3,4,5]. *T. equi* induces disease characterized predominantly by fever, hemolytic anemia, and signs associated with erythrocyte lysis [2].

Recent clinical studies with *T. haneyi* have characterized clinical disease in spleen-intact horses. These studies suggest that *T. haneyi* causes minimal morbidity in this population, similar to what is frequently reported for *T. equi*, in which horses show no overt clinical signs without careful observation. Unfortunately, clinical signs induced experimentally are not always consistent with those observed outside of a controlled environment, especially when animals are used for work and/or athletic purposes [2,6,7]. Indeed, there is significant global variation in disease severity, and the parasite- and host-specific factors associated with variations in virulence are not well understood. The lack of understanding of *Theileria* pathogenesis, in particular factors leading to significant morbidity and mortality, is a primary reason many countries restrict entry of horses infected with *T. equi*. Both species are capable of long-term, persistent infection in spleen-intact horses and consequently provide a continual, silent reservoir for transmission [1,2,8].

Beyond infection dynamic studies, comparative genomic analysis of *T. equi* and *T. haneyi* has revealed that *T. haneyi* has undergone genomic reduction. Specifically, the genome of *T. haneyi* is approximately 2 Mbp smaller than that of *T. equi* [9]. The consequences of genomic reduction are not yet known, with the exception of the loss of susceptibility to imidocarb diproprionate (ID) [10]. In other organisms, there are mixed data as to whether genomic reduction is associated with an increase or decrease in virulence [11,12].

Virulence is defined as “the relative capacity of a microbe to cause damage”, and in order to characterize virulence for a pathogen, it must be compared to a standard [13]. Previous authors have used rate of mortality, a composite of quantifiable parameters related to morbidity (anemia, fever, and weight loss) or parasite-specific parameters (peak parasitemia, duration of parasitemia, and time to peak) as parameters for comparison [13,14,15,16]. Comparisons are often made between a wild-type strain and a genetically modified strain; however, in this case, genetic modification was provided through evolution of *T. haneyi,* and *T. equi* was used as the standard [13]. Even though the two species are phylogenetically distinct, they do share similar pathogeneses, further enabling comparison [1].

Splenectomy is often used within equine babesiosis and theileriosis research to assess the efficacy of chemotherapeutics in eliminating persistent infection [1,10,17,18,19]. This technique is utilized because splenectomy decreases the host’s capacity to control parasitemia, thereby enabling large-scale expansion of previously undetectable parasite populations to detectable, and sometimes clinically significant, levels [20,21]. Within splenectomized horses, both *T. equi* and *T. haneyi* induce overt clinical disease, but, unlike *T. haneyi*, in the absence of intervention, *T. equi* infection almost always results in fatal hemolytic anemia with perimortem hemoglobinuria and icterus [1,10,22,23,24]. The mortality rate and clinical course of *T. haneyi* infection of splenectomized horses were not well characterized prior to this study.

For this study, we define virulence as a composite of mortality rate, peak parasitemia, and percent packed cell volume decline in splenectomized horses. The objective of this study was to utilize clinical data from nine splenectomized, *T. haneyi*-infected horses to determine if *T. haneyi* is less virulent than *T. equi*. These data were compared to previous published studies concerning the outcome of splenectomy on *T. equi*-infected horses [1,10,22,23,24].

## 2. Results

### 2.1. Characterization of Experimental T. haneyi Infection of Splenectomized Horses

Following inoculation, all horses in Group 1 (Table 1; naïve splenectomized) developed a fever during acute infection, which included the first 60 days post-inoculation. Parasites were first observed via light microscopy between days 7 and 21. Peak fever ranged from 102.4 to 105.8 °F and correlated with rising parasitemia or peak parasitemia. Parasitemia appeared in waves over the course of infection. Horses 248 and 344 had more significant increases in parasitemia than that observed in 285 and 301 (Figure 1).

For horses inoculated via whole transfusion, the first peak in parasitemia occurred between 12 and 15 days post-inoculation, with the single horse inoculated with stabilates peaking the first time at 23 dpi (Figure 1). The second peak occurred for transfused horses at 29–32 dpi, followed by a third wave in the surviving horses at 43–50 dpi (Table 2). Horse 285 did develop additional parasitemia spikes over the course of infection at 54 and 69 dpi (Table 2, Figure 1). Overall, horses developed significant changes in the packed cell volume/hematocrit (PCV/HCT), with the decline ranging from 33.33 to 72.22% (Table 2). HCT/PCV nadirs were greatest for horses 248 and 344, both of whom were infected via whole-blood transfusion from an acutely infected spleen-intact horse. Within Group 1, only a single horse, 248, was euthanized.

### 2.2. Characterization of Experimental T. haneyi Infection of Spleen-Intact Horses

Horses in Group 2 (splenectomized after the development of persistent infection) developed mild, transient clinical signs during acute infection with *T. haneyi,* as described in [10], and subsequently entered a state of persistent, asymptomatic infection. Following splenectomy, all developed a fever with peak rectal temperature ranging from 101.2 to 103.8 °F. Each horse in this group exhibited a single peak in parasitemia that was equal to or greater than that observed in the horses in Group 1 (Figure 2). All four horses experienced peak percent parasitized erythrocytes (PPE) between days 14 and 19 post-splenectomy, with HCT nadirs at 6–18 days after the peak (Table 3). The percent decline in HCT was similar to that observed in Group 1 and ranged from 33.52 to 76.53%.

### 2.3. Characterization of Complete Blood Count Parameters in Experimentally Infected Horses

For horse 344 (splenectomized prior to infection; Group 1) and for all horses in Group 2, additional monitoring of CBCs enabled greater assessment of leukocyte and platelet dynamics over the course of acute infection (Table 3). Horse 344 exhibited significant changes in leukocyte and platelet counts during the acute period. The horse developed a lymphocytosis and monocytosis, along with periods of neutropenia and thrombocytopenia (Table 3, Figure 2). For Group 2 horses, only two experienced a brief period of neutropenia within the first 3 days post-splenectomy. All horses in this group developed neutrophilia, monocytosis, and thrombocytopenia (Table 3 and Figure 3). Three out of four horses also developed a lymphocytosis. The monocytosis observed in Group 2 horses was typically present when parasites were observed by light microscopy, with the elevation declining as parasites declined (Figure 3). This trend occurred in 3 out of 4 horses and for 344 (Figure 2 and Figure 3). Horse 275 exhibited only a brief period of monocytosis (Figure 3). All horses, however, did develop monocyte peaks after peak parasitemia was reached and peaks occurred either on the same day or prior to anemia nadirs (Table 3 and Figure 3). Monocyte peaks ranged from 1.03 to 11.44 × 10^3^/µL of whole blood with the normal range being 0.2–0.6 × 10^3^/µL of whole blood (Figure 2 and Figure 3). Each horse also experienced periods of thrombocytopenia around the time(s) of peak parasitemia (Figure 4 and Figure 5). Prolonged clotting times were not appreciated. All Group 2 horses experienced intermittent periods of neutrophilia, whereas 344, the single horse splenectomized prior to infection, (344) experienced periods of neutropenia (Table 3).

### 2.4. Monitoring of T. haneyi Parasitemia via nPCR and Blood Smear Cytology

Following acute infection, all horses except horse 248 transitioned to state of chronic, asymptomatic infection. Horses were serially monitored by both light microscopy and *T. haneyi* nPCR to assess whether parasitemia remained detectable. For horses in Group 1, parasites were occasionally detected by light microscopy and horses occasionally became mildly anemic. Two of the three horses that survived acute infection remained nPCR positive. Each of these horses was euthanized due to the development of either chronic renal failure or acute guttural pouch empyema with concurrent severe laminitis. Due to the low number of animals in this study, it is unclear whether the intercurrent diseases that manifested during the persistent phase of infection are related to debilitation caused by infection with *T. haneyi*. The final surviving horse, 344, remained nPCR positive for approximately 11 months following initial infection. Horse 344 subsequently maintained a parasitemia level below the limits of detection by *T. haneyi* nPCR for subsequent 40 months. All horses in Group 2 were positive for *T. haneyi* for a minimum of 106 days post-splenectomy via nPCR, and two of the four horses maintained a parasitemia level below detectable limits of nPCR for the subsequent 10 months (Table 4). Horse 278 was nPCR negative for 6 months prior to testing weak positive and then negative at all subsequent timepoints.

## 3. Discussion

In this study, clinical data from naïve, splenectomized horses infected with *T. haneyi,* and from horses persistently infected with *T. haneyi* that subsequently underwent splenectomy, were retrospectively assessed to generate a clear picture of the pathogenesis of *T. haneyi* and of its virulence in splenectomized horses compared to *T. equi.* Previous studies have confirmed the necessary role of the spleen in controlling *T. equi* parasitemia, and splenic removal invariably leads to uncontrolled parasite replication and usually mortality [2]. The spleen alone is not sufficient to reduce morbidity as studies in foals with severe combined immunodeficiency (SCID) revealed adaptive immunity is also crucial for parasite control [25]. While the exact immunologic mechanism(s) of hemoparasite control in equine theileriosis have not been elucidated, studies in other hemoparasitic diseases have revealed that removal of parasitized erythrocytes by the reticuloendothelial system of the spleen is integral to parasite control. In this, cells of histiocytic lineage remove parasitized erythrocytes, which are either opsonized by antibody or demonstrate reduced deformability, from circulation and destroy them, thereby reducing the circulating parasite burden [26].

Significantly, in this study, the overall survival rate of *T. haneyi*-infected, splenectomized horses was 87.5%. This is in stark contrast to the high fatality of *T. equi* in splenectomized horses, which ranges from 75 to 100% [17,18,19,22,23,24,27]. Peak parasitemia for all splenectomized, *T. haneyi*-infected horses in this study was less than 15% and decline in PCV/HCT ranged from 33.52% to 76.53%. *T. equi*-infected, splenectomized foals in one study developed peak parasitemias of >37%, and a percent decline in PCV of 73% or greater [23]. In one study using splenectomized donkeys, PPE reached 80%, and all eight donkeys died within 7–9 days [22]. Multiple studies assessing the efficacy of chemotherapeutic drugs also evaluated splenectomized horses. In these studies, untreated splenectomized control horses infected with *T. equi* developed parasitemias as high as 48%, with PCV declines of >70% of pre-inoculation levels [18,19]. Overall, infection with *T. equi* leads to a consistently greater peak parasitemia and a greater decline in PCV, resulting in hemoglobinuria, icterus, weakness, and exercise intolerance. None of the *T. haneyi*-infected, splenectomized horses in this study developed hemoglobinuria or icterus, and the horses experienced only brief, intermittent periods of weakness and exercise intolerance. Thus, objective, clinical comparison of *T. haneyi* and *T. equi* infection of splenectomized horses clearly demonstrates that *T. haneyi* is less virulent.

Interestingly, whereas naïve, splenectomized horses infected with *T. haneyi* developed multiple parasitemia peaks and associated PCV/HCT nadirs, splenectomy of horses after they had reached the persistent, asymptomatic phase of infection led to recrudescence characterized by only a single parasitemia peak and associated, transient anemia. Chronic infection of horse prior to splenectomy would have enabled the development of robust, adaptive immunity, including responses to antigenic escape variants, should they exist, which may have contributed to the single peak observed. The difference in parasite dynamics is in agreement with previous studies in *T. equi* that suggest that adaptive immunity is a significant contributing component of immunologic control [25]. Antibody responses to *T. equi* have been shown to correlate with the transition to persistent infection [28].

This study also revealed that a proportion of splenectomized, *T. haneyi*-infected horses in both Groups 1 and 2 eventually reached a state of undetectable parasitemia consistent with apparent clearance. This decline is in stark contrast as to what has been documented with *T. equi* recrudescence following splenectomy of persistently infected horses, wherein splenectomy following infection leads to recrudescence and often death [10,18,19]. Little is yet known regarding how equids transition from acute, symptomatic theileriosis to asymptomatic theileriosis. IgG responses in adult horses appear to correlate with control [28]. In malaria, chemotherapeutic treatment can lead to a period of dormancy [29]; however, the reported period of dormancy is short compared to the duration of undetectable parasitemia observed in these horses following splenectomy (up to 40 months), and undetectable parasitemia was observed in horses that never underwent imidocarb dipropionate treatment. Repeated exposure and resultant boosting of the adaptive immune response to some hemoparasitic diseases likely also enable maintenance of an asymptomatic state [30]. More recently, the concept of disease tolerance has been recognized, wherein a single exposure to an organism leads to host adaptations that minimize the inflammatory response yielding clinical disease [31]. This sort of adaptation could play a role in the observed, reduced disease severity in *T. haneyi.* Furthermore, it is possible that removal of the spleen sufficiently increased exposure of the *T. haneyi* parasites to the adaptive immune system, enabling enhanced, targeted control after a period of long-term exposure. To this end, the reduced virulence of *T. haneyi* is critical in that the infected, splenectomized host equids survived long enough for this degree of immune control to develop. Given that a greater proportion of horses in Group 2 cleared infection than those in Group 1, one additional possibility is that previous exposure to *T. equi* led to a cross-protective immune response that facilitated clearance of *T. haneyi.* Additional studies are required to elucidate the definitive mechanism of *T. haneyi* clearance in splenectomized horses.

Close evaluation of the hematologic changes observed in *T. haneyi*-infected, splenectomized horses also provided a window into the cellular (leukocyte and platelet) dynamics at play during *T. haneyi* infection. With each of the peaks of parasitemia for both groups of horses, hematocrit or packed cell volume and platelet counts subsequently declined substantially. For both groups of horses, HCT/PCV declined to nadirs within 0–18 days of the parasitemia peak, with time to decline related to the height of the parasitemia peak, i.e., the larger the peak, the less time to the HCT/PCV nadir. Furthermore, for horses in Group 2, nadirs occurred within 8 days for 3 out of 4 horses. The outlier, horse 275, was mildly dehydrated despite efforts to encourage drinking. Thus, his total protein and HCT were artificially elevated. The decline in HCT/PCV following peak parasitemia, with more rapid decline occurring after higher parasitemia peaks, is similar to that observed in horses infected with *T. equi* [18,19,23], and is due to both erythrocyte lysis following parasite replication and clearance of infected erythrocytes from circulation [2].

The development of thrombocytopenia overlapped with both significant declines in HCT and periods of peak parasitemia, and is also observed in *T. equi*-infected horses. The mechanism of thrombocytopenia occurring in *Theileria* sp. is not yet known; however, based on work in other systems, including *Plasmodium* sp., several potential mechanisms have been proposed, including immune-mediated platelet destruction, excess consumption following vasculitis and/or endothelial damage, and decreased production [2,32].

In recent years, malaria researchers have ascertained an additional mechanism for thrombocytopenia that is relevant in cases in which the platelet count is low but the patient lacks evidence of coagulopathy (as in *T. haneyi* and *T. equi*). Kho et al. found that platelets form complexes with *Plasmodium* sp.-infected red blood cells (iRBCs) and are subsequently not counted by machines. This study also demonstrated that platelets directly contribute to the elimination/killing of the erythrocytic stage of *Plasmodium* spp. [33]. This mechanism could explain the apparent normal clotting times for all Group 2 horses and for horse 344 despite profound thrombocytopenia. Furthermore, periods of thrombocytopenia appear to precede nadirs in PCV/HCT of all horses, and correlate with peaks of parasitemia, suggesting possible involvement of platelets with parasite control.

Neutrophilia and lymphocytosis were also observed in the single Group 1 horse for which data were available (344) and for all Group 2 horses. A neutrophilia has also been observed in both *T. equi* infection of splenectomized and spleen-intact horses [24,34,35]. However, those research groups noted a lymphopenia in conjunction with the neutrophilia in splenectomized horses [24,34]. A study of acute *T. equi* infection in field cases in Chile did note occasional lymphopenia or lymphocytosis in the cohort of cases evaluated [35]. Neutrophilia and lymphopenia have also been frequently noted in human patients with malaria, and the lymphopenia frequently transitions to a lymphocytosis over time [36,37]. As neutrophils are among the first responders of the innate immune response, elevations in neutrophil count were expected with changes in parasitemia [37]. Lymphopenia may occur in spleen-intact horses due to stress-induced apoptosis and reallocation to sites of inflammation with rebound lymphocytosis due to redistribution of lymphocytes [36]. In our splenectomized cohort, an additional contributing factor to the lymphocytosis may be splenectomy, as multiple studies of splenectomized humans have noted lymphocytosis post-splenectomy, with redistribution to the peripheral circulation being a proposed mechanism [38,39].

The marked monocytosis in *T. haneyi*-infected, splenectomized horses has not been reported outside of the publication describing the original experiment for which horses in Group 2 were splenectomized [10]. The significant changes in monocyte count immediately follows peak parasitemia, and may be a compensatory response for the removal of iRBCs [40,41]. This presumption stems from what has been elucidated from infection of spleen-intact humans with malaria. Specifically, red pulp macrophages continually remove senescent and less malleable erythrocytes (including *P. falciparum*-infected erythrocytes) that are incapable of passing through the inter-endothelial slits of the red pulp [42]. *Theileria* iRBCs are suspected to be more rigid due to biochemical changes induced induced by intraerythrocytic infection, and this increased rigidity has been documented in malaria [22]. These changes in RBC deformability subsequently lead to increased clearance of these damaged RBCs in the spleen [42,43]. However, in splenectomized horses, this major contributor to erythrocyte clearance and resultant decreased severity of disease, is absent. Therefore, with the loss of red pulp macrophages (RPM), other phagocytic cells, such as monocytes and neutrophils, may contribute more to phagocytosis and removal of infected cells from circulation, especially during acute infection [44,45]. Kim specifically showed an increased frequency of monocytes in RPM-deficient, *Plasmodium*-infected mice that appear to serve a similar role to RPMs in their absence [44]. Sponaas, additionally, found that increased monocyte populations contribute to control of blood-stage malaria [40]. Outside of the experimental setting, monocytosis has also been observed in both symptomatic and asymptomatic acute cases of human malaria [44,46,47,48,49].

An alternative perspective is that monocytes may also contribute to pathology. Other research groups have shown that monocytes contribute to severe anemia and dysregulated cytokine secretion in malaria [50,51]. Our group found that during acute, lethal infection of cattle with the related parasite, *Theileria parva*, animals that recover from infection exhibit a distinctly different monocyte phenotype than animals that succumb to infection, suggesting that expansion of intermediate and non-classical monocyte phenotypes is a correlate of severe disease in *Theileria* [52]. Ultimately, additional studies are warranted to assess changes in monocyte subsets and function during acute *T. equi* and *T. haneyi* infection to elucidate whether they potentiate protection or pathology.

## 4. Materials and Methods

### 4.1. Equine Infection Studies

Retrospective analyses were performed using data from horses previously utilized in equine theileriosis studies between October 2011 and October 2019 [1,8,10]. All experiments were approved by the University of Idaho and/or Washington State University IACUCs (ASAFs 2013-66, 2016-18, 2016-28, and 6241). Records from naïve, splenectomized horses infected with *T. haneyi* between 2011 and 2016, and from persistently infected horses that were subsequently splenectomized for verification of chemotherapeutic clearance (July–October 2019), were utilized (Table 1) [1,8,10,17,18,19]. Horses were divided into two groups based on the timing of splenectomy. Horses in Group 1 were splenectomized prior to *T. haneyi* infection and then inoculated 27–392 days later either through whole-blood transfusion or via intravenous inoculation with infected blood erythrocyte stabilate. The inoculum for all horses in this study was developed from the original splenectomized horse used to amplify *T. haneyi* parasitemia for genome sequencing (horse 208) [1]. Following inoculation, horses were clinically monitored as previously described [9,10]. Two additional horses, 285 and 344, were also included in this evaluation. The horses were inoculated via whole-blood transfusion from a previously infected, spleen-intact horse (344) or via stabilate inoculation (285) (Table 1). Both horses were monitored daily for changes in parasitemia, fever, and PCV/HCT. Horse 344 was additionally monitored through serial complete blood counts (CBCs) and serum biochemical profiles. Group 2 horses were co-infected with *T. equi* and *T. haneyi* as described in [8,10]. Briefly, horse 280 was infected with *T. haneyi* first, and clinically monitored until a state of asymptomatic, persistent infection, characterized by normal complete blood count and physical exam despite positive *T. haneyi* nPCR, was reached. At that point, 280 was super-infected with *T. equi*, and again monitored until a state of asymptomatic, persistent infection was reached. Using the same monitoring parameters, horses 275, 277 and 278 were infected with *T. equi* first, and *T. haneyi* second. The horses then underwent two rounds of imidocarb dipropionate treatment separated by 2–4 months. Horses were splenectomized 657–743 days later (between 2022 and 2474 days after infection). All horses utilized in this study were negative for *T. equi* via nPCR at all timepoints post-imidocarb dipriopionate therapy and following splenectomy, but remained *T. haneyi* positive.

### 4.2. Blood Collection

Blood was collected via jugular venipuncture daily during the initial acute infection period and serially thereafter for verification of persistent infection. Blood was collected into both ethylenediaminetetraacetic acid (EDTA) and serum separator tubes during both the acute and chronic phases of infection. Blood was processed within two hours of collection with 100 µL aliquots frozen for subsequent DNA isolation. The serum separator tubes were allowed to clot and then subsequently centrifuged to enable serum separation. Serum was then utilized within 2 h for assessment of serum biochemical profiles or aliquoted and stored at −20 °C.

### 4.3. Light Microscopy

During the acute infection, Giemsa- or Diff-Quik-stained blood smears were evaluated daily for changes in parasitemia. Parasitemia was calculated as previously described [10].

### 4.4. T. haneyi Nested PCR

DNA was isolated from previously aliquoted whole-blood samples that had been stored at −20 °C. Following thawing at room temperature for 5 min, the DNeasy Blood and Tissue Kit (Qiagen, Inc., Venlo, The Netherlands) was utilized for isolation per the manufacturer’s instructions. DNA was subsequently utilized immediately for PCR or frozen at −20 °C until analysis. [10] The primers utilized amplify a *T. haneyi*-specific gene that is absent from the *T. equi* genome and the assay was performed as previously described [8,10].

## 5. Conclusions

In summary, retrospective assessment of clinical data from *T. haneyi*-infected, splenectomized horses utilized in previous studies from our lab demonstrates that *T. haneyi* is less virulent than *T. equi,* as evidenced by the comparative reduction in mortality and percent PCV/HCT decline in splenectomized, infected horses. This is also the first report of apparent, spontaneous clearance of *T. haneyi* by splenectomized horses, which, to the author’s knowledge, has not been observed in *T. equi*. Moreover, this analysis provides insight into the potential role of innate immune effector cells in combatting acute infection that is more readily apparent when the spleen (and its red pulp macrophages) is removed. Splenectomy of persistently infected horses (Group 2) also revealed that long-term exposure to *T. haneyi* and the resultant memory immune response may enable more rapid control of parasitemia during subsequent periods of recrudescence. Further evaluation is warranted to better characterize this adaptive immune response. Additionally, further evaluation of monocytes and platelets is warranted to determine if and how they are involved in parasite clearance and/or immunopathology during acute and persistent equine theileriosis. In future studies, the discrepant virulence characteristics of these two closely related species of *Theileria* may be exploited via comparative genomic, transcriptomic, and proteomic analyses during acute and persistent infection to dissect mechanisms of virulence and long-term persistence in *Theileria.*

## Figures and Tables

**Figure 1 pathogens-11-00254-f001:**
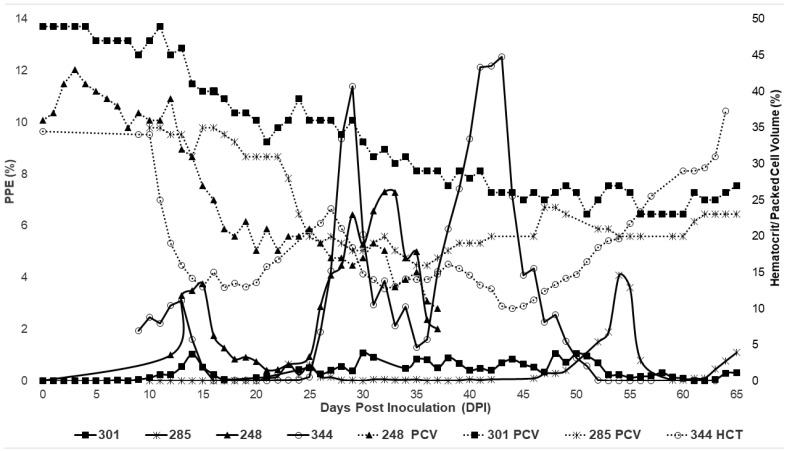
Changes in parasitemia (solid lines) and hematocrit (dotted lines) during acute *T. haneyi* infection of splenectomized (Group 1) horses.

**Figure 2 pathogens-11-00254-f002:**
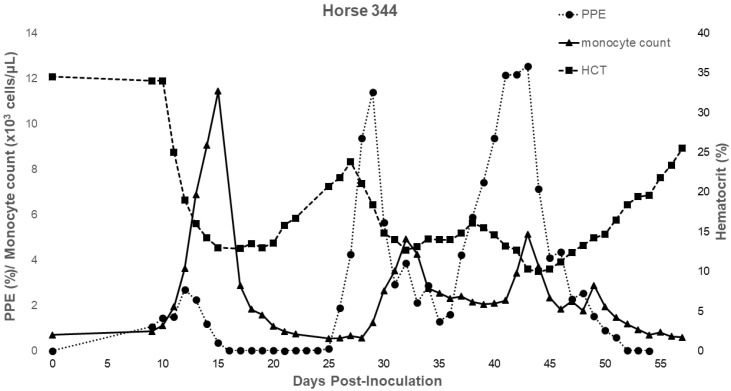
Changes in parasitemia, monocyte count, and hematocrit during acute *T. haneyi* infection of a splenectomized horse (344) in Group 1.

**Figure 3 pathogens-11-00254-f003:**
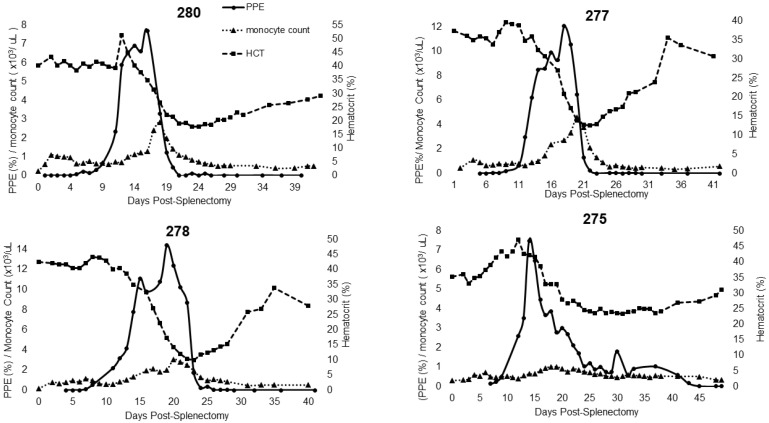
Changes in parasitemia, monocyte count, and hematocrit during acute recrudescence of *T. haneyi* post-splenetomy in persistently infected (Group 2) horses.

**Figure 4 pathogens-11-00254-f004:**
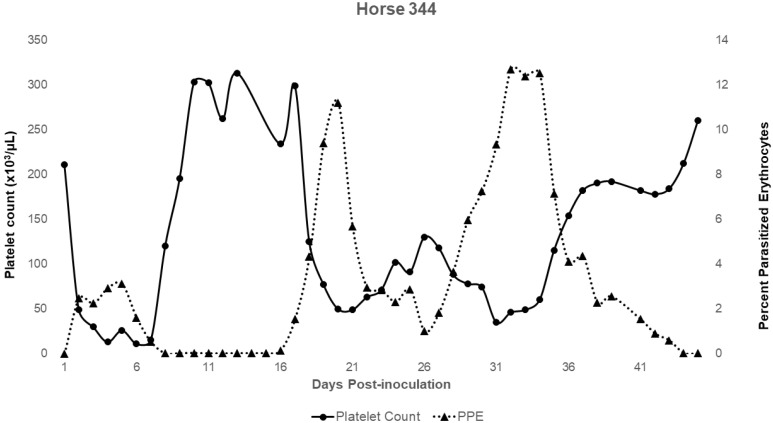
Changes in platelet count during acute *T. haneyi* infection of a splenectomized horse (344) in Group 1.

**Figure 5 pathogens-11-00254-f005:**
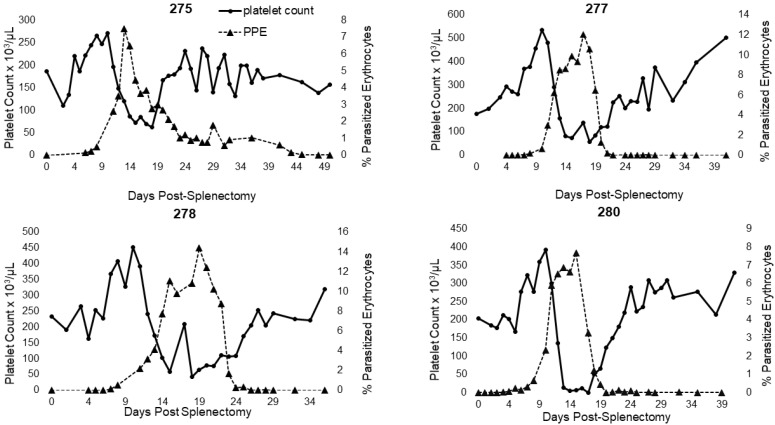
Changes in platelet count during acute recrudescence of *T. haneyi* post-splenectomy in persistently infected (Group 2) horses.

**Table 1 pathogens-11-00254-t001:** Inoculation methodology and timing of splenectomy for study horses. PPE = percent parasitized erythrocytes.

Horse	Group	Inoculation Type	Stabilate Volume and PPE	Splenectomy Pre- or Post-Inoculation	Associated Study
248	1	Blood—120 mL	2 mL/12% (donor)	Pre	[1]
301	1	Blood—120 mL	2 mL/12% (donor)	Pre	[1]
285	1	Stabilate	5 mL/8.8%	Pre	This study
344	1	Blood—150 mL	4 mL/8.8% (donor)	Pre	This study
275	2	Stabilate	2 mL/12%	Post	[8,10]
277	2	Stabilate	2 mL/12%	Post	[8,10]
278	2	Stabilate	2 mL/12%	Post	[8,10]
280	2	Stabilate	2 mL/12%	Post	[8,10]

**Table 2 pathogens-11-00254-t002:** Changes in parasitemia and packed cell volume during acute infection of horses in Group 1. PPE = percent parasitized erythrocytes; PCV = packed cell volume.

Horse	248	285	301	344
PPE peaks	15 dpi—3.75% 32 dpi—7.32%	23 dpi—0.65% 54 dpi—4.1% 69 dpi—2.34%	14 dpi—1.03% 30 dpi—1.08% 50 dpi—1.06%	13 dpi—3.10% 29 dpi—11.40% 43 dpi—12.55%
PCV nadirs	20 dpi—18% 37 dpi—10%	29 dpi—18% 54 dpi—20% 72 dpi—15%	21 dpi—33% 38 dpi—27% 51 dpi—23%	15—13% 32—12.7% 44—10%
Maximum% change in PCV ^#^	−72.22%	−57.14% ^#^	−53.06%	−71.01%
Anemia duration	15–37 dpi	23–87 dpi	35–67 dpi	11–62
Survival	Euthanized 37 dpi	survived	survived	Survived

^#^ PCV change is based upon PCV recorded on day 10, which was the first recorded value for this horse.

**Table 3 pathogens-11-00254-t003:** Hemogram changes in 344 (Group 1) and Group 2 horses post-splenectomy.

Hemogram Changes	344 (dpi)	275 (dps)	277 (dps)	278 (dps)	280 (dps)
Monocytosis	11–22, 30–57	15–28	14–24	13–28	15–26
Lymphocytosis	11–67	None	15–27, 29	19, 21–31	20–24,27
Lymphopenia	None	1–3	None	None	None
Neutropenia	30–39, 42–51	1–3	None	None	2
Neutrophilia	28	7, 9–10	3, 6–10, 19–25	6–29	9–11, 13–14, 20–21, 23–30
Anemia duration	11–62	20–42	17–32	18–34	18–43
Anemia nadir	15, 32, 44	31	22	24	24
Maximum% change in HCT	−71.01%	−33.52%	−66.31%	−76.53%	−55.72%
Parasitemia peak(s)	13, 29, 43	14	17	19	17

**Table 4 pathogens-11-00254-t004:** Long-term outcome and associated nPCR results for all horses.

Horse	Survival Acute Infection	Long-Term Survival	nPCR 6–12 Months Post-Infection/Splenectomy
248	N	N	Positive—37 dpi
301	Y	Euthanized; chronic renal failure	Positive—1402 dpi
285	Y	Euthanized; guttural pouch empyema and acute, severe laminitis	Positive—556 dpi
344	Y	Y	No longer detected; last positive 267 days post-inoculation; negative at 350 dpi and last negative at 1565 dpi
275	Y	Y	Positive; 536 days post-splenectomy
277	Y	Y	No longer detected after 106 dps (1/17/20), then neg on 3/3, 5/2, and 7/24, faint pos on 9/23, then neg 11/17, and 1/5/21
278	Y	Y	No longer detected after 152 dps
280	Y	Y	No longer detected after 171 dps

## Data Availability

All data can be found in the manuscript.

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
