# Peer review of "Clinical Progression of Theileria haneyi in Splenectomized Horses Reveals Decreased Virulence Compared to Theileria equi"

_pathogens, 2022, doi:10.3390/pathogens11020254_

Round 1

Reviewer 1 Report

Sears et al. presented an excellent study titled Clinical development of Theileria haneyi in splenectomized horses indicates lower virulence compared to Theileria equi. Both Theileria equi and Theileria haneyi are Apicomplexan hemoparasites that cause horse theileriosis. This study provided new information regarding Theileria haneyi infection in horses, which previously had very little information. Additionally, this study defines pathogenicity in splenectomized horses as a composite of mortality rate, peak parasitemia, and percent packed cell volume loss. Additionally, it examines clinical data from nine splenectomized T. haneyi-infected horses to establish if T. haneyi is less pathogenic than T. equi.

Further research is required to fully understand the mechanism of Theileria haneyi, particularly the immune system of the host against this parasite.

Author Response

We thank the reviewer for their close review of our findings and kind comments.

Reviewer 2 Report

The current manuscript by Sears et al. undertook a comparative approach between two Theileria species, T. haneyi and T. equi, in horses. The pathogenesis and importance of T. haneyi as a novel emerging microorganism should be studied and this work provides further information on the clinical picture and long-term outcome of T. haneyi infection.

To me the general conclusion that T. haneyi is less virulent than T. equi due to less mortality rate and other clinical criteria (parasitaemia, drop in PCV and etc.) seems enough justified however the second conclusion which is spontaneous clearance of T. haneyi in splenectomized horses remains relatively obscure and a major point of concern. This conclusion is entirely based on PCR results of G2 group where all horses except 275 where negative for T. haneyi 6-8 month post-splenectomy. In contrast G1 animals that lacked spleen at the time of infection remained PCR positive up to 3-4 years, so obviously in G1 there is no clearance of T. haneyi. In addition, 3 out 4 animals in G2 have already experienced a T. equi infection (section 4.1), thus it seems that in this group the immune response to T. equi (a closely related species) is probably affecting persistence of T. haneyi and that potentially cross-species immunity exists and contributes to clearance of T. haneyi in less than a year. In addition, G2 animals survived better without any long-term complications seen in G1 horses that necessitated euthanasia. I would rather change this conclusion or discuss the possible effects of a past T. equi infection on T. haneyi infection outcome in G2.

Minor comments:

  • What does PPE stands for?
  • Line 210: apparence clearance or apparent clearance?
  • Line 283: Theileria

  • In materials and methods a section or reference on splenectomy is missing.

  • The timing of parasite inoculation and splenectomy is not precisely described for G1 and G2. e.g. how many days post-inoculation the spleen where removed in G2? how many days post-splenectomy the animals were infected with the parasite?

  • As far as I understood, G2 animals were splenectomized during when persistent T. haneyi was established, however in all graphs (Fig. 3 and Fig. 5) were PPE of G2 animals is presented the PPE is 0 at day 0? Or the parasitaemias were so low that can not be seen on the current graph scales.

  • How was parasitaemia trends of G2 animals prior to splenectomy and upon infection with T. haneyi?

Author Response

We thank the reviewer for the thoughtful and detailed review of our manuscript.  Please find our responses to each individual critique below.  

To me the general conclusion that T. haneyi is less virulent than T. equi due to less mortality rate and other clinical criteria (parasitaemia, drop in PCV and etc.) seems enough justified however the second conclusion which is spontaneous clearance of T. haneyi in splenectomized horses remains relatively obscure and a major point of concern. This conclusion is entirely based on PCR results of G2 group where all horses except 275 where negative for T. haneyi 6-8 month post-splenectomy. In contrast G1 animals that lacked spleen at the time of infection remained PCR positive up to 3-4 years, so obviously in G1 there is no clearance of T. haneyi. In addition, 3 out 4 animals in G2 have already experienced a T. equi infection (section 4.1), thus it seems that in this group the immune response to T. equi (a closely related species) is probably affecting persistence of T. haneyi and that potentially cross-species immunity exists and contributes to clearance of T. haneyi in less than a year. In addition, G2 animals survived better without any long-term complications seen in G1 horses that necessitated euthanasia. I would rather change this conclusion or discuss the possible effects of a past T. equi infection on T. haneyi infection outcome in G2.  

We thank the reviewer for this input.  We agree that the definitive meaning of the apparent clearance of T. haneyi is unclear.  For that reason, we did not emphasize the finding in the title or the abstract.  To clarify, 1/4 horses in group 1 cleared infection, and 3/4 horses in group 2 cleared infection, so it it was not only horses in group 2 that cleared the infection.  It is interesting that more animals in group 2 cleared the parasite, and there are many variables that could be in play -- including immune response development, interactions of the two Theileria sp., genetic variability between horses, etc.  We have expanded the discussion of this topic to include the reviewer's theory, and others, of how/why clearance occurs, and to emphasize that more work is needed to clarify this potential feature of T. haneyi infection in the future. 

Regarding the one horse in group 1, 248, that was euthanized due to T. haneyi infection (and not intercurrent disease):  This was the first horse we had ever infected with this isolate, and we were expecting the isolate to behave more similarly to T. equi.  Thus, we euthanized the horse when the PCV/hematocrit dropped to 10%.  The horse, however, did not develop any clinical signs attributable to severe, acute hemolysis that we see in T. equi (hemoglobinuria, icterus), and was more mild.  Our IACUC protocols at this time mandated euthanasia due to the PCV drop.  However, these protocols were subsequently amended and other horses (e.g. horse 344) reached similar PCV nadirs and survived to become asymptomatic.  Furthermore, following splenectomy, some horses in group 2 developed similar PCV nadirs to 248 and 344.  Thus, we are not comfortable  making any conclusions on severity differences due to  T. equi exposure.    

  • What does PPE stands for?  It stands for percent parasitized erythrocytes.  Thank you for catching this.  We have added this definition in line 112 in the text (the first time it is mentioned in text) and have also defined it in the table legends.
  • Line 210: apparence clearance or apparent clearance?  Thank you for catching this.  It is meant to be 'apparent clearance' and has been fixed.
  • Line 283: Theileria  Thank you for catching this.  It has been fixed.
  • In materials and methods a section or reference on splenectomy is missing.  Thank you for catching this.  We have added the appropriate references in the text.
  • The timing of parasite inoculation and splenectomy is not precisely described for G1 and G2. e.g. how many days post-inoculation the spleen where removed in G2? how many days post-splenectomy the animals were infected with the parasite?  Thank you; we have added this information into the text in the methods (lines 322 and 339).
  • As far as I understood, G2 animals were splenectomized during when persistent T. haneyi was established, however in all graphs (Fig. 3 and Fig. 5) were PPE of G2 animals is presented the PPE is 0 at day 0? Or the parasitaemias were so low that can not be seen on the current graph scales.  Figures 3 and 5 depict changes in PPE and hematocrit or platelet count following splenectomy.  The X axis is the days post splenectomy, so day 0 is day 0 after splenectomy.  At that point, the PPE was too low to detect (e.g. no parasites detected on blood smear), but it quickly increased again, and the hematocrit and platelets dropped -- this is all due to recrudescence of the parasite.  We have double checked that all graph axes are labelled clearly.
  • How was parasitaemia trends of G2 animals prior to splenectomy and upon infection with T. haneyi?  This is discussed in references 8 and 10 for these specific horses.  The clinical signs were extremely mild, with only mild decreases in hematocrit, and a brief period of fever.  All horses were completely normal by day 60 following infection.  We have double-checked that these citations are clearly noted in the text so that readers with similar questions may find the information.